# Urban Bird Community Assembly Mechanisms and Driving Factors in University Campuses in Nanjing, China

**DOI:** 10.3390/ani13040673

**Published:** 2023-02-15

**Authors:** Zixi Zhao, Amaël Borzée, Jinghao Li, Sheng Chen, Hui Shi, Yong Zhang

**Affiliations:** 1Co-Innovation Center for Sustainable Forestry in Southern China, College of Biology and the Environment, Nanjing Forestry University, Nanjing 210037, China; 2Laboratory of Animal Behaviour and Conservation, College of Biology and the Environment, Nanjing Forestry University, Nanjing 210037, China; 3Jiangsu Agricultural Biodiversity Cultivation and Utilization Research Center, Nanjing 210014, China; 4Center for Biological Disaster Prevention and Control, National Forestry and Grassland Administration, Shenyang 110141, China

**Keywords:** urban greenspaces, bird community assembly, landscape metrics, diversity, urban planning

## Abstract

**Simple Summary:**

Accelerated urbanization has changed the composition of regional landscape patterns, directly affecting the composition of bird communities. This study analyzes bird community assembly mechanisms and the driving factors in university campuses in Nanjing, China. We found that the phylogeny of bird communities in all universities followed a pattern of aggregation. Grass, water, and buildings were the main factors affecting the campus bird communities’ functional and phylogenetic diversity. Based on our results, we offer several practical measures for urban planners to better protect urban biodiversity and develop eco-friendly cities.

**Abstract:**

University campuses are important components of cities, harboring the majority of urban biodiversity. In this study, based on monthly bird survey data covering 12 university campuses located either downtown or in the newly developed areas in Nanjing, China, in 2019, we studied the assembly processes of each campus’s bird population and their main drivers by modeling a set of ecological and landscape determinants. Our results showed that (1) bird abundance and species diversity in the newly developed areas were significantly higher than in those downtown; (2) the phylogeny of bird communities in all universities followed a pattern of aggregation, indicating that environmental filtering played a major role in community assembly; (3) specifically, grass, water, and buildings were the main factors affecting each campus’s bird community’s functional and phylogenetic diversity, with the areas of grass and water habitats having a significant positive correlation with phylogenetic diversity, while the size of building areas was negatively correlated. Our results emphasize that habitat features play a decisive role in determining urban bird population diversity and community assembly processes. We suggest that increasing landscape diversity, e.g., by reasonably arranging the location and area of water bodies and grasslands and improving the landscape connectivity, could be a powerful way to maintain and promote urban bird diversity.

## 1. Introduction

The Earth is undergoing its sixth mass extinction [1]. Numerous driving factors, operating at different trophic levels, both directly and indirectly trigger mass extinctions, including habitat loss and destruction, climate change, illegal harvesting, environmental pollution, and invasion by alien species. Among them, habitat loss and degradation as a consequence of human activities play a profound role in influencing global biodiversity [2]. Urbanization processes, in particular, transforming rural, semi-natural, and natural habitats into highly impervious and fragmented land surfaces, are increasingly critical in determining the distribution of wildlife [3], and hence, their community composition and assembly processes.

The assembly processes of wildlife in natural and semi-natural habitats have been extensively studied; environmental filtering and competitive exclusion are the main mechanisms determining the wildlife’s community assembly processes [4]. The environmental filtering hypothesis suggests that species coexisting can adapt to the current environment and that there will be convergence and the aggregation of functional traits [5]. The competitive exclusion hypothesis explains that similar species aggregating in a community will compete for the environment’s limited resources [6]. With the increasing intensity of competition, the subordinate species will be outcompeted, weakening the species similarity of the community [7]. However, because of limitations in the niche theory, the coexistence of species in highly diverse ecosystems is not explained. Hence, the neutral theory, which takes ecological drift, diffusion restriction, immigration, and emigration into account, highlights the importance of random processes [8]. Thus, the niche theory and neutral theory are compatible and complementary, jointly explaining the assembly processes of wildlife from different angles [7]. Studies on the mechanisms of community assembly processes are mainly focused on natural habitats [9], and those of wildlife relying on urban habitats are largely unexplored. Research is still needed to better understand the mechanisms governing urban wildlife assembly processes, especially at different spatial and temporal scales [10].

Birds are among the most studied taxonomic groups in biodiversity research as they are widely distributed, easily detected, and sensitive to environmental changes [11]. Urban birds rely on urban green spaces as their main habitat, including urban parks, green spaces, and street trees, which provide suitable feeding, perching, and breeding conditions. However, in opposition to the natural environment, urban habitats are intensively influenced by anthropogenic activities and land-use policies; hence, the underlying mechanisms shaping urban biodiversity patterns are miscellaneous and rather complicated. As a result, additional research is critical to better understand and safeguard urban biodiversity [12].

The size of urban green spaces is commonly recognized as one of the main factors affecting bird diversity, along with other variables, such as water sources and human interference [13]. In addition, the plant community structure of urban green spaces is positively correlated with bird diversity [14], and natural water sources will encourage the presence of urban birds [15]. Artificial green spaces and water sources can also contribute to maintaining urban bird diversity by providing suitable foraging and breeding habitats [16]. Conversely, human interference, such as habitat fragmentation, increases in impervious surfaces, and noise may lead to a decline in bird species’ richness and abundance [15].

Moreover, bird communities may respond differently to man-made environments. For example, a study in 17 cities in ten European countries found that bird abundance during the breeding season was positively correlated with the available area of green space but was negatively affected by building area [17]. However, another study found that phylogenetic diversity increased with an increase in building area, impervious surface, and pedestrian flow density [18]. Therefore, to comprehensively understand the drivers of urban bird community composition, it is essential to evaluate the effects of the urban environment on multiple dimensions of diversity.

As an important component of urban habitats, university campuses host abundant biodiversity [19]. Campuses with plenty of trees and shrubs can provide excellent feeding, breeding, and nesting sites for a considerable proportion of bird species [20]. For instance, the Shannon–Wiener and Simpson biodiversity indices of campus birds increased with increasing areas of green space and water and decreased with an increase in the fragmentation degree of landscape patches (split index) [19]. Moreover, urbanization may also lead to homogeneous bird communities as only a few species are able to adapt to the new environment [21] and some species are likely to have vanished when the original landscape was urbanized [22].

Hence, in this study, to explore the effect of urban expansion on bird assembly processes, based on monthly bird survey data in 2019 covering 12 university campuses located downtown or in the newly developed areas of Nanjing, we first compared the bird species diversities of the downtown and newly developed areas. We predicted that the bird species diversity downtown would be lower than that of newly developed areas, where the greenspace area is larger and is less fragmentated. We then determined the bird assembly processes by examining both the functional and phylogenetic diversity. To understand the community assembly mechanisms, we analyzed the landscape attributes of the study area through the interpretation of remote sensing images and used linear mixed models to determine the main variables affecting bird diversity. Our results thus provide practical guidelines for maintaining biodiversity and will inform urban campus planning.

## 2. Materials and Methods

### 2.1. Study Area

Nanjing (31°14′–32°37′ N, 118°22′–119°14′ E), the capital city of Jiangsu Province, is located in the central region of the lower Yangtze River Reaches in eastern China [23]. Nanjing is one of the most important central cities in China, with a total population of 9.31 million and a total area of 6587.02 km^2^. As of 2014, the urbanization rate of Nanjing had reached 80.92%. In recent years, many universities have moved their campuses to the suburbs as population growth put a strain on urban space, thereby offering a good opportunity to study the effects of urbanization on bird diversity [24]. We selected 12 university campuses, with six located downtown and the other six in the newly developed area (Figure 1).

### 2.2. Data Collection

#### 2.2.1. Landscape Configuration

We used high-resolution Google Earth images (version) (Landsat8, Mountain View, CA, USA, with a spatial resolution of 4.51 m) captured in December 2018 to classify the land use on each campus. We acquired vector maps delineating the boundaries of the universities under study from each university’s official website and overlaid them onto the Google Earth image. We then classified the land-cover types of each university campus through visual interpretation and on-the-spot verification. In total, the landscape was divided into five categories: buildings, grass (such as lawns and grassed areas), forest (including trees and shrubs), impervious surfaces (such as parking lots, playgrounds, and squares), and water (including ponds, streams, and urban lakes).

We used ArcGIS v.10.8 (Environmental Systems Research Institute, RedLands, CA, USA) to calculate the area (in km^2^) of each landscape type within each university campus. In addition, we calculated the total length (km) of roads on each campus, using the computational geometry tools available in ArcGIS. Then, we used FRAGSTATS v.4.2.1 (Department of Forest Science, Oregon State University, Corvallis, OR, USA) to calculate the fragmentation indices at both patch and landscape levels for all campuses. In total, eight landscape indices were selected, including patch density (PD), largest patch index (LPI), edge density (ED), mean patch area (Marea), the proportion of similar adjacencies (PLADJ), connectance index (CI), aggregation index (AI), and splitting index (SPLIT).

#### 2.2.2. Bird Counts

We conducted monthly bird surveys from January to December 2019. In each campus, we set up two line transects covering various habitat types, including water, grass, buildings, forest, and impervious surface. The lengths of the transects were dependent on campus size. The shortest transect was 0.51 km, while the longest transect was 1.75 km, with an average length of 0.96 ± 0.33 km (mean ± SD). During the surveys, observers walked along the transects at a speed of ca. 2 km/h and recorded the species and number of birds that were either seen visually or that were heard. Birds flying from the opposite direction to the observers were recorded, while those moving from back to front were not counted. When a large group of birds was encountered, we used a batch-counting method [25].

#### 2.2.3. Diversity Metrics of Bird Assemblage

We divided the year into four seasons, comprising spring (March to May), summer (June to August), autumn (September to November), and winter (December to February). We calculated the functional diversity (FD) using the traits collected from the life history and ecological characteristics of 1205 species of Chinese birds [26]. We used three continuous variables, including body size (mm), dispersal ratio (ratio of mean wing length to the cube root of body weight), clutch size (n), and five discrete variables: nest site, nest type (cave, tree, shrub, water, or ground-nesting), hunting vulnerability (never or seldom killed, occasionally hunted, or killed), migration patterns (resident, part of the population is resident, or completely migrant), and habitat specificity (the number of habitat types for a species). We first calculated the functional distance of each bird species using Gower’s distance, and then applied the unweighted pair group method with arithmetic mean (UPGMA) to reconstruct dendrograms [27].

The strength of phylogenetic signals in functional traits is a major factor determined through the use of phylogenetic diversity to estimate the ecosystem function of a species [28]. Hence, we calculated four indices, namely, the mean pairwise distance (MPD), the mean nearest taxon distance (MNTD), and their standardized values, SESMPD and SESMNTD, respectively. The indices of SESMPD and SESMNTD are commonly used to determine the community assembly rules [29]. If the SES value is greater than 0, this indicates that the community assembly process is mainly generated by overdispersion, while a value of *SES* lower than 0 suggests that the community assembly process mostly relies on clustering [30]. We used the formula below for calculations:(1)$$SES=\frac{\mathrm{Obs}-{Mean}_{null}}{{SD}_{null}}$$
where Obs is the observed diversity value, *Mean_null_* is the mean value of the randomized assemblages, and *SD_null_* is the standard deviation of the randomized assemblages [31].

### 2.3. Statistical Analyses

First, we compared the differences in landscape indices between newly developed and downtown areas, using an independent sample *t*-test as our data, following the normal distribution. Then, we compared the number of bird species (bird species richness) and the functional and phylogenetic diversity between the newly developed and downtown areas among the seasons. We tested whether there were significant differences from the linear regression models, followed by Tukey post hoc tests. We examined the relationships between bird species richness and standardized functional and phylogenetic diversities using Pearson’s correlations, after checking the data normality. All analyses were conducted using SPSS v.26 (IBM SPSS; Chicago, IL, USA).

We standardized the calculated functional diversity (also SESMPD and SESMNTD), phylogenetic diversity (also SESMPD and SESMNTD), and the independent variables using Z-scores. Then, we applied linear mixed models to detect the effects of landscape metrics on the functional diversity and phylogenetic diversity of campus birds. We used the bird diversity indices as dependent variables and landscape metrics as independent variables (Table 1). Before that, we calculated the variance inflation factor (VIF) to detect collinearity problems, and variables with VIF values larger than 4 were removed. We used the Akaike information criterion for small sample sizes (AICc) to select the best models. All models with a delta AICc value of less than 2 were included in the best model set. Then, model averaging was applied to acquire the average parameter estimation for each variable. The explanatory powers of the landscape variables regarding bird diversities were also measured by averaging the values of Adj.r2, calculated based on the optimal models [32]. All analyses were conducted using the ‘FD’, ‘MuMIn’, ‘picante’, and ‘vegan’ packages in R4.1.1 [33,34].

## 3. Results

### 3.1. Landscape Metrics

Our results showed that the building areas downtown were significantly larger than those of the newly developed area. There were significant differences in mean patch area between the newly developed and downtown areas (df = 10, t = −2.939, *p* = 0.015). The mean patch area (0.17 ± 0.03, mean ± SD) in the downtown area was significantly lower than that in the newly developed area (0.28 ± 0.09). The proportion of similar adjacency (0.17 ± 0.03) in the downtown area was significantly lower than that in the newly developed area (0.28 ± 0.09, df = 10, t = −4.611, *p* = 0.002). The forest area was the most significant land type in all the university campuses studied (Appendix A). NU2 had the highest proportion of forest area (82.94%) while SEU1 had the lowest proportion of forest area (25.45%; see Table 1 for abbreviations). Water area was the smallest landscape type. The campus with the largest proportion of water area was MCMU2 (6.28%) and there were no water bodies in NU1. SEU1 had the highest proportion of building area (29.18%) while the campus with the lowest proportion of building area was NNU2 (10.88%).

### 3.2. Bird Community Composition, Assembly Processes, and Biodiversity

A total of 3842 birds belonging to 42 species were recorded at the campuses located downtown. In newly developed areas, a total of 7674 birds belonging to 58 bird species were recorded. Analyzing bird residential types showed that a total of 41 species were resident birds (62.1%), 12 species were winter migrants (18.1%), 10 species were summer migrants (15.1%), and the remaining 4.5% were passengers (Appendix A).

### 3.3. Bird Communities Functional and Phylogenetic Diversity

The standardized functional diversities (SESMPD and SESMNTD) of bird communities for six of the universities were greater than zero, indicating that the bird communities there resulted from overdispersion, including three universities in the newly developed areas (NCMU2, NFEU2, and NNU2) and three in the downtown areas (NFU1, NJU1, and NAU1). The standardized phylogenetic diversities (SESMPD and SESMNTD) of the bird communities in all 12 universities were all less than zero, indicating that the bird communities resulted from clustering (Figure 2).

The functional and phylogenetic diversity of bird communities at the university campuses in the newly developed area was significantly higher than at the campuses downtown (Figure 3). The bird community functional and phylogenetic diversity also differed among seasons. In detail, the significant differences in functional diversity (F = 31.898, *p* < 0.001) only occurred in spring and summer (Figure 3), while the significant differences in phylogenetic diversity (F = 30.820, *p* < 0.001) were detected in summer and autumn (Figure 3).

We found a significantly positive relationship between species richness and functional diversity in spring (Figure 4), while the relationships between species richness and functional diversity based on the data collected during the three other seasons and the whole year were weak. The annual species richness was significantly correlated with phylogenetic diversity (Figure 4), while the relationships between seasons were not significant.

After examining the collinearity, five independent variables were retained, namely, building area, the connectance index, water, grass, and the splitting index. Based on the results of the linear mixed models and the model averaging, building, water, and grass areas were always included in the best model, explaining the functional and phylogenetic diversity of campus birds (Figure 5; Appendix A). For the full year, two variables (water and building area) featured in the best-predicting model for FDMPD (adj.r^2^ = 0.24, weight = 0.71), with only building area being significantly and positively correlated. We found a significant positive effect of grass on functional diversity in spring, with an explanatory power of 0.93 and a model weight of 0.94. In summer, water area was a significant negative predictor for FDMPD (adj.r^2^ = 0.25, weight = 0.39). Building, grass, and water areas were significant predictors of phylogenetic diversity in summer, being negatively correlated with building area and positively correlated with the others (adj.r^2^ = 0.92, weight = 0.55). In winter, we found a significant negative effect of building area on PDMNTD (adj.r^2^ = 0.05, weight = 0.43; Figure 5; Appendix A).

## 4. Discussion

Our results showed that bird communities in urban campuses were the result of overdispersion and neutral assembly, according to the functional diversity hypothesis. However, when examining the phylogenetic diversity, bird communities in all campuses were clustered, supporting the environmental filtering hypothesis in the community assembly processes, where the influence of the abiotic environmental factor resulted in species with similar functional traits aggregated in the same community. Landscape metrics, such as grass, water, and building areas were the most critical variables in determining the multiple dimensions of campus bird diversity, highlighting the importance of landscape diversity in maintaining and promoting urban biodiversity.

### 4.1. Landscape Metrics

We found the mean patch area and the proportion of similar adjacency of downtown areas to be smaller than the newly developed areas, suggesting that the degree of fragmentation was higher downtown [22]. Since campuses downtown comprised dense residential buildings, and scattered grassland and forest areas, the landscape was more highly fragmented [14]. Compared with the downtown areas, the landscape of newly developed areas was more closely connected with a larger proportion of greenspace.

### 4.2. Bird Diversity and Residential Types

We found that the proportion of resident birds was the largest, followed by migratory birds. The proportion of passengers was the smallest. This result suggested that university campuses can offer good habitat conditions for many urban bird species throughout their life history [13]. In addition, migratory birds can also use campus habitats for part of their life history. Analyzing the The International Union for Conservation of Nature threatened species categories, we found that all the birds recorded in this study were listed as Least Concern (Appendix A). Human disturbances may partly explain this finding, as university campuses often have a higher human population. In addition, the local species pool also affected our results as only a few rare species were recorded in our study area.

### 4.3. Differences in Bird Functional and Phylogenetic Diversity in Newly Developed and Downtown Areas

We found the functional and phylogenetic diversity of birds in newly developed areas to be higher than in the downtown areas. Urban birds are generally expected to be negatively affected by increasing urbanization [35]. Newly developed areas, being away from the city center, allow universities to harbor a high proportion of forest and water areas, while campuses located downtown have a long architectural history, and their proximity to downtown businesses is not supportive of a green environment [20]. Hence, habitats in the newly developed areas are more suitable for resident urban birds to live and breed, and these habitats have also become an energy supply station for migratory birds. For instance, we found summer migratory birds, such as the striated heron (*Butorides striata*), and winter migratory birds, such as the Eastern spot-billed duck (*Anas zonorhyncha*) in NCMU2 and NNU2 in the newly developed area. This result is consistent with previous findings in Mexico, where a significant difference in bird diversity between the New North Area of Mexico and the transitional area of Mexico emphasized that urban green space made an important contribution to the diversity of migratory birds. Similar results were also found in Michigan [36]. Although the urban landscape area is relatively small, it can also provide significant protective value for birds when combined with the surrounding landscape matrix [37].

We found that bird species richness was positively correlated with functional diversity in spring and phylogenetic diversity for the whole year. During the spring migration, some birds may utilize urban habitats as stopover sites. With the increase in species richness, the functional traits of birds in the community could also become diverse, and the genetic relationship among species could change accordingly [38]. Our findings are also supported by an earlier study in Italy, suggesting that as the number of birds decreased, the species richness and functional diversity became positively correlated [39]. Due to functional redundancy, we expect that this mathematical correlation will weaken as the number of species in the community increases [27].

The results of the functional diversity analyses showed that the bird community was overdispersed at six campuses (NFU1; NJU1; NAU1; NCMU2; NFEU2; NNU2), while it was randomly structured at the other campuses (Figure 2), indicating that interspecific competition played a major role in the process of bird community structure. Bird species with similar functional traits often share the same resources [40]. This result is also in line with the limiting similarity hypothesis, wherein interspecific competition is the main driver determining the community structures by limiting the similarity of species’ body size [41]. Coexisting bird species in assemblages that vary in body size reduce the overlap in resource use, contributing to higher functional diversity [42].

The results of the phylogenetic diversity (PDMPD and PDMNTD) in all university campuses were less than zero, suggesting that bird assemblages were characterized by a clustered community structure (SES value of between −2.5 and −1.0), following the environmental filtering hypothesis [43]. Habitat environments on campuses are relatively simple, which can atract bird species with similar ecological requirements in terms of habitat and food demands, resulting in closely related species being assembled in the same community [44]. Moreover, habitat fragmentation may also play a role. University campuses often contain a large proportion of buildings and playgrounds [45], resulting in a highly fragmented landscape. Many birds have specific criteria to select their habitat and are highly dependent on their environment; as a result, these criteria act as strong habitat-filtering variables [46]. Therefore, in the process of community assembly, the effect of environmental filtering is stronger than that of interspecific competition [47].

### 4.4. Relationship between Landscape Metrics and Diversity

Building, water, and grass areas were consistently included in the best models, predicting multiple dimensions of bird diversities in the different seasons for the whole year. The results of the model average showed that the effects of the three landscape variables on bird diversities differed among the seasons (Figure 5). This may be explained by the complex life history of birds as their ecological requirements vary across life history stages and seasonality [48]. We found a significant negative effect of building areas on phylogenetic diversity, especially in summer. The building area positively correlated with phylogenetic diversity and the explanatory power for this variable was also the highest [49]. This result indicates that with an increase in building area, the available habitat decreases, along with food resources; hence, interspecific competition among species that share the same ecological niche increases [50]. The risk of collision with buildings also hinders the spread of species [51]. Moreover, the mortality rate of migratory birds is 90% higher than that of the local resident birds, which greatly limits the phylogenetic diversity of birds [52].

Functional diversity increased with grass and water areas, especially in spring and winter. One reason is that vegetation and water sources in university campuses provide suitable perching and breeding places for resident species [20]. Compared with natural habitats, urban green space plays an important role for resident birds [53] and helps maintain ecosystem functions and the resilience of avian communities [17]. Grass areas are one of the most important habitats of urban birds, and mowing forbs have a higher insect abundance and diversity, offering abundant food resources for insectivorous birds, while seeds are foraged by granivorous and omnivorous birds. In addition, artificial light at night close to grasslands can attract phototactic insects, which will increase the abundance and diversity of omnivorous birds and, thus, the overall gradient of the species community and bird heterogeneity [17]. Similarly, water areas can offer suitable habitats for water-bird species and drinking resources for urban bird species. Hence, campuses with a larger proportion of water area are effective in maintaining higher bird diversities.

## 5. Conclusions

Our results showed that the phylogenetic structure of bird communities in our study area was mainly affected by environmental filtering, highlighting a set of landscape attributes that affect multiple dimensions of bird diversity. Specifically, building area had a negative impact on the phylogenetic diversity, while water and grass had a positive impact. In order to better protect urban bird communities and promote urban bird diversity, urban planners need to design continuous large areas with water bodies and grasslands. We should not only provide a stable habitat and breeding sites for resident birds but also provide appropriate footholds for migrating birds to satisfy their energy requirements. Therefore, in future urban planning, minimizing habitat fragmentation and providing large areas of grassland and water bodies for birds could play a positive role in greening the environment and, hence, maintaining and promoting biodiversity in urban environments.

## Figures and Tables

**Figure 1 animals-13-00673-f001:**
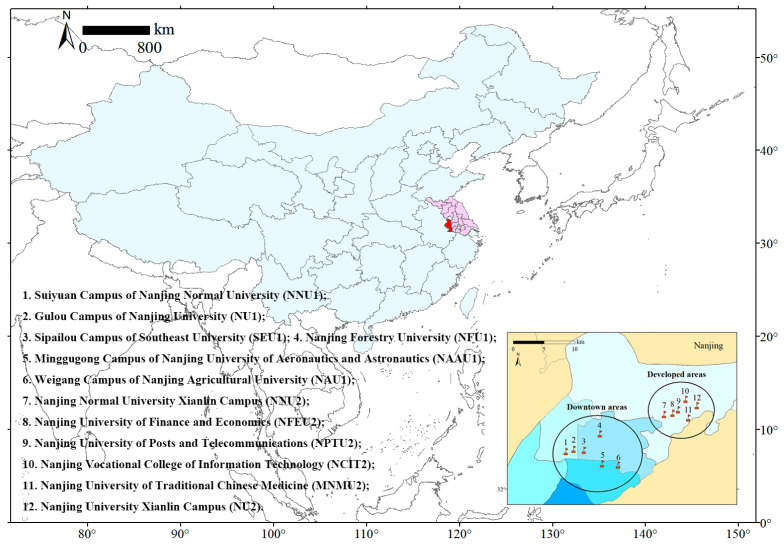
The geographic location of Nanjing in China and the distribution of the 12 university campuses studied for this paper.

**Figure 2 animals-13-00673-f002:**
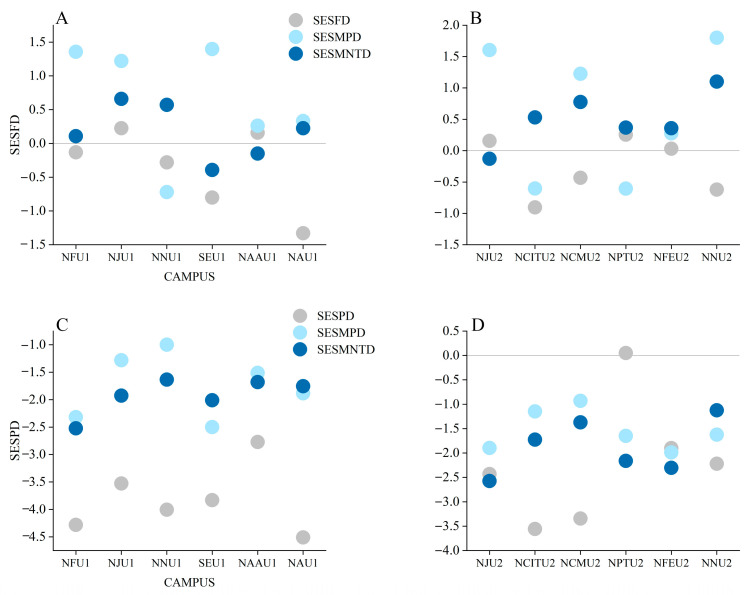
Standardized functional and phylogenetic diversities at 12 university campuses. (**A**,**C**), bird diversities at the campuses located in downtown areas; (**B**,**D**), bird diversities at the campuses in the newly developed areas. SESFD: standardized values of functional diversity; SESPD: standardized values of phylogenetic diversity; SESMPD in (**A**,**B**): standardized values of the mean pairwise distance of functional diversity; SESMNTD in (**A**,**B**): standardized values of the mean pairwise distance of functional diversity; SESMPD in (**C**,**D**): standardized values of the mean nearest taxon distance of phylogenetic diversity; SESMNTD in (**C**,**D**): standardized values of the mean pairwise distance of phylogenetic diversity. For abbreviations of each campus name, see Figure 1.

**Figure 3 animals-13-00673-f003:**
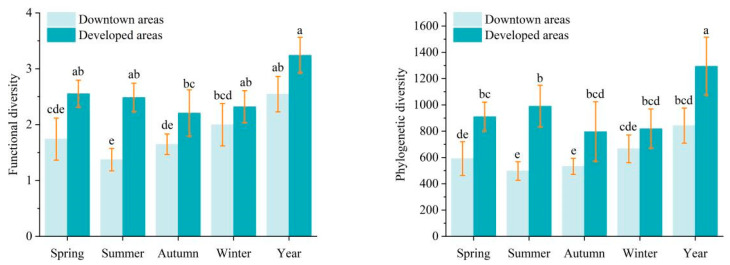
Comparing the annual and seasonal bird functional and phylogenetic diversity at the campuses located in the downtown and newly developed areas in Nanjing. Different letters above the bars indicate statistical significance at the 0.05 level, measured by Tukey’s post hoc test.

**Figure 4 animals-13-00673-f004:**
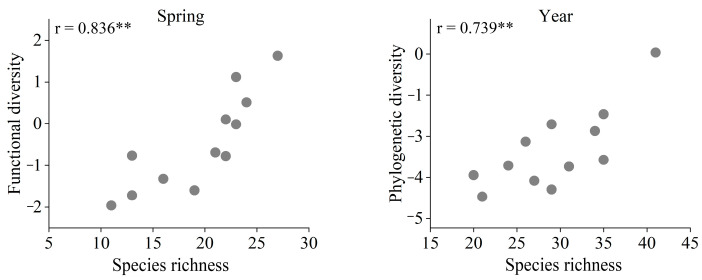
The relationships between species richness and the functional and phylogenetic diversity of bird communities in spring and over the whole year. The strength of the relationship is expressed as Pearson’s correlation coefficient, r (** *p* < 0.01). Non-significant relationships are not shown in this figure.

**Figure 5 animals-13-00673-f005:**
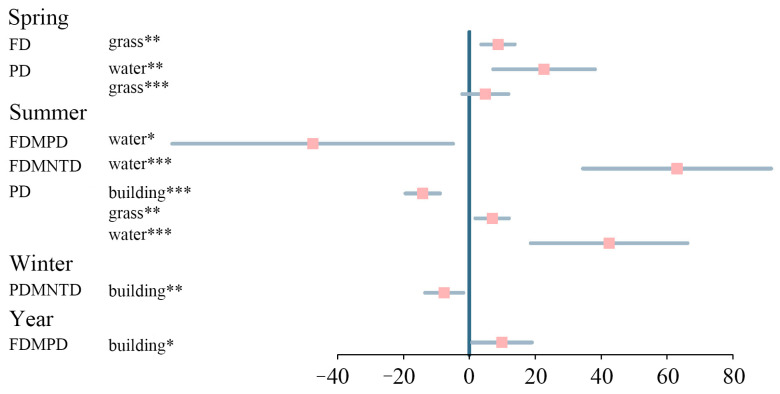
The average parameter estimates (standardized regression coefficients) of variables featured in the best models predicting bird diversities in different seasons (only significant variables are shown; for the others, see Appendix A). FD: functional diversity; PD: phylogenetic diversity; FDMPD: mean pairwise distance of functional diversity; FDMNTD: mean pairwise distance of functional diversity; PDMNTD: mean pairwise distance of phylogenetic diversity. The *p*-value of each independent variable is given as * < 0.05; ** < 0.01; *** < 0.001. For definitions of the abbreviations of the different variables, see Table 1.

**Table 1 animals-13-00673-t001:** Description of the landscape metrics and the abbreviations used in this study.

Landscape Metrics	Abbreviations	Brief Description
Total area	Ta	The total area of the campus
Forest area	Forest	Woody plant area including trees and shrubs
Grass area	Grass	Grass and lawn area
Building area	Building	Building and structure area
Impervious surface area	Isurf	Impervious surface area, including parking areas, playgrounds, and squares
Water body area	Water	Water body area, including ponds, rivers, and urban lakes
Patch density	Pd	The density of a patch in the landscape
Largest patch index	Lpi	Dominant patch types in the landscape
Edge density	Ed	Landscape fragmentation degree
Mean patch area	Marea	Average area of a patch in the landscape
Proportion of similar adjacency	Pladj	Patch proximity and landscape fragmentation
Connectance index	Connect	Patch connectivity
Splitting index	Split	Separation degree of patch distribution
Aggregation index	Ai	Connectivity between patches of landscape types

## Data Availability

The data that support the findings of this study are available from the corresponding author upon reasonable request.

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
