# Peer review of "Urban Bird Community Assembly Mechanisms and Driving Factors in University Campuses in Nanjing, China"

_animals, 2023, doi:10.3390/ani13040673_

Round 1
Reviewer 1 Report
Please see attached comments

Author Response
Comment 1
The manuscript will benefit by adding more structure. I suggest to separate the Results and Discussion sections into sub-headings. Possible suggestions for the Results are Land-use, Bird abundance and richness, Bird communities and seasons (or Functional diversity and seasons, Phylogenetic diversity and seasons), Bird communities and land use (or Functional diversity and land use, Phylogenetic diversity and land use), etc. The same sub-sections should be used in the Discussion, to make it easier to follow ideas and the relevance and importance of each result.
>>Thank you for your thoughtful comments. Followed your suggestion, we have modified results and discussion sections and added appropriate subtitles accordingly.
Comment 2
Using FRAGSTATS the authors calculated 8 fragmentation indices at patch and landscape levels for all campuses. It would be interesting to know what the study found in relation to habitat fragmentation. Please see the More detailed comments, number 8.
>>Thank you for your comments so much. This part has been modified accompany with comment 8.
Comment 3
Considering how much data was collected from 12 University campuses, the discussion of the results and recommendations to maintain biodiversity seem quite general. Specially since the value of the study, as stated by the authors themselves, is that the results provide practical guidelines to maintain biodiversity and guide urban campus planning. Therefore, please include in the Discussion, a section about bird abundance/richness/functional/phylogenetic diversity at specific named campuses, with reference to particular bird orders/species that were found in the study. Did any of the campuses host important breeding/wintering/migrating populations of birds, any rare or endangered species?
>>Thank you for your comments so much. We have added a paragraph in the discussion part (4.2 Bird diversity and residential types) to summary the resident types of birds investigated, geographic area and IUCN red list information encountered in this study.
Comment 4
Please also include in the discussion a section with specific recommendations and suggestions on how the campuses that were studied may maintain and/or improve biodiversity for particular bird species/groups. Did any of the campuses stand out as an excellent example of urban planning? Were any of the campuses doing everything wrong?
>>Thank you for your comments. We think it is a good direction to study. However, as this manuscript mainly focuses on bird assembly processes and biodiversity, analyses for particular bird species/group is out of the scope of this study, we are happy study this issue in more detail later on.
Comment 5
The manuscript needs to be corrected by a native English-speaker as there are many language errors and information is not clear at some points. Extensive editing of the English language and style are required.
>> The manuscript has been gone through by a native speaker.
MORE DETAILED COMMENTS
1.Simple Summary, Line 14: The summary should start with 1 or 2 sentences explaining what the study was about. It is confusing for the reader to immediately start reading about the results with no information about the context of the study.
>>Thank you for the comment. The background information of this study has been included.
- Abstract, Lines 25-31: Most of the abstract includes background information. The background information should be shortened. The abstract should also include the most important results and conclusions of the study.
>>Suggestion token over, the abstract part has been revised.
- Introduction, Lines 102-104: It is stated that ‘We predicted that the bird species diversities downtown would be lower than that of newly developed areas’. Why? What was this based on? Please include an explanation clarifying the differences between downtown and newly developed areas.
>>Thank you for your comments. We have justified our prediction in more details.
- Introduction, Lines 104-105: It is stated that ‘We then determined the bird assembly processes by examining both functional and phylogenetic diversities.’ Did you have any predictions for this point, similarly to the previous one? If yes, please include them, with an explanation.
>>Thank you for your comments. Above, Bird species diversities include both functional and phylogenetic diversities, and the predictions are the same. Here, we use functional and phylogenetic diversities to explore bird assembly processes.
- Lines 119-120: It is stated that ‘We selected 12 university campuses, six located downtown and the other six in the newly developed area (Fig 1).’ See More detailed comments, number 3 above.
>>Thank you for your suggestion. In combination with the comment 3 above, we have explained the differences between the newly developed and downtown urban areas in the revised version.
- Lines 125 & 135: Move Landscape configuration to 2.2.1 and Bird counts to 2.2.2
>>Done.
- Bird counts: Either include a map for each campus, showing the habitats and transects or a Table showing the number of transects and total transect lengths per campus.
>>Thank you for your suggestion. We have added ‘Table S2 in the Supplementary Materials to show the transect information.
- Lines 204-210: Fig S1 and Fig S2 present the classificaton of land use type at all University campuses. These are interesting figures. Please also include a quantitative comparison of habitat types and levels of habitat fragmentation between downtown and newly developed areas, specially since the hypotheses of this study is based on the assumption that these areas are different from each other. Please explain how different they are, in relation to the availability of natural habitats, and the extent of habitat fragmentation. Are these differences significant?
>>Thank you for your suggestion. We now used T-test to detect the differences of the landscape metrics between the old and newly development urban areas as our data followed the assumption of normality.
- Lines 211-213: Please include a Table with a breakdown of bird abundance and species richness at each University campus, as well as info on how many are resident, migratory and breeding species. It would also be interesting to know what bird orders were found at each campus, and whether there were any rare/endangered species.
>>Thank you for your nice suggestion. We have added 'Table S3' to the Supplementary Materials, a list of bird species in 12 campuses of Nanjing, the residence type, geographical area, and the IUCN red list threated classes are also listed.
- Lines 284-287: It is stated that ‘Newly developed areas being away from the city center allow universities to harbor a high proportion of forest and water areas, while campuses located downtown have a long architectural history and their proximity to downtown businesses is not supportive to a green environment.’ Were the differences in the availability of natural habitat and of habitat fragmentation significantly different between downtown and newly developed areas, in this study? Please explain. Also, see More detailed comments, numbers 3 and 5 above.
>>Thank you for your suggestion. Please see the responses to your comments, 3 and 5 above.
LANGUAGE AND OTHER MINOR CORRECTIONS
- Line 17: replace with ‘…. buildings …’
>>changed
- Line 25: replace with ‘… of avian community assembly processes …’
>>Done.
- Line 54: replace with ‘….. similar species aggregating in a community …’
>>Done.
- Line 81: replace with ‘… habitat fragmentation …’
>>Done.
- Line 82: include a citation for the study
>>Done.
- Lines 86-88:‘However, another study found that phylogenetic diversity increased with increasing building area, impervious surface, and pedestrian flow density.’Is number 18 the citation for this study? If yes, move it to the end of this sentence.
>>Done.
Lines 95-96: It is not clear what is meant here. Do you mean that the Shannon-Wiener and Simpson indices of campus birds decreased with decreasing diversity of landscape patches? Please rephrase
>>Thank you for your suggestion. We have revised the sentence to make it more clearly.
- Line 116: it should be km2
>>Changed
- Lines 116-118: It is stated that‘As of 2014, the urbanization rate of Nanjing has reached 80.92%.’What does urbanization rate refer to? Since Nanjing is a city, isn’t it 100% urban? Please clarify.
>>Although Nanjing is a huge city, it is still not fully urbanized. Same as Shanghai, although it is the largest city in China, there still have country area.
- Line 137: do you mean a spatial resolution of 4.51m? If yes, please include ‘spatial’
>>Done.
- Line 146: it should be km2
>>Changed.
- Line 147: delete ‘also’
>>Done.
- Line 147: delete ‘the road length (km) of each campus’ and rephrase as ‘the total length of roads in each campus’
>>Suggestion taken over.
- Line 181: I am assuming that with ‘bird species richness’ you are referring to the number of bird species recorded. Please include the definition.
>>Yes, thanks, we have revised the sentence to avoid confusion.
- In Line 142, grass is defined as ‘lawns and grass’ and in Table 1 it is defined as grassland area. Please use the same definition throughout the manuscript.
>>We have changed the description in table 1.
- Lines 212-213: rephrase as ‘In newly developed areas, a total of 7674 birds belonging to 58 bird species were recorded.’
>>Suggestion taken over.
- Line 229: Replace Table 1 with Figure 1
>>Done.
- Figure 3. Please indicate on the graphs, either with bold or an asterisk, the
statistically significant comparisons.
>>Thank you for your suggestion. We tried to use asterisk to show the differences, but as there are so many bars, the figure looks very confused, so we chose to keep using letters.
- Line 251: replace ‘connecting’ with ‘connectance index’
>>Done.
- Line 277: rephrase with ‘where the influence of the abiotic environment results in species with similar functional traits being selected.’
>>Done.
- Lines 290 & 291: species names should be in italics
>>Changed.
- Lines 312-314: It is stated that ‘However, comparing with the natural ecosystem, campuses provide less resource for birds, and hence interspecific competition happens’. It is not clear what is meant here. Please rephrase or delete.
>>Thank you for your suggestion. The sentence has been deleted.
Reviewer 2 Report
Review
In the conditions of the onset of the period of "the sixth great mass extinction" on planet Earth, the reviewed study is relevant. So far, research is at the stage of searching for answers to the question: what elements of the landscape are favorable for the formation of a diverse community of birds? However, in the near future, humanity, while transforming natural landscapes, will be forced to build artificial landscapes (including cities) in such a way as to not only ensure the existence of the human population, but also conservation the species diversity. The authors have obtained interesting results that allow to give recommendations to architects of the integrated development of settlements already today.
I have no comment on the research methodology, data analysis, presentation of results, illustrations, discussion and conclusions.
There is only one important note to research design: this is a one-year study. Annual changes in bird populations and environmental factors have a large impact on the bird community. If the patterns identified by the authors were confirmed in other years, it would be great.
The remaining comments are minor, the authors will be able to quickly make changes to the manuscript.
Abstract
In my opinion, it is necessary to describe in more detail not only what has been done, but also the results obtained and the patterns identified, as, for example, it is written in Simple Summary
Introduction
lines 86-87: “However, another study found that phylogenetic diversity increased with increasing building area, impervious surface, and pedestrian flow density” [need references].
lines 90-91: “As an important component of urban habitats, university campuses host abundant biodiversity globally”. The use of the word "globally" raises doubts. Need references too.
Line 96: “the Shannon-Wiener and Simpson indices” need corrected to “the Shannon-Wiener and Simpson biodiversity indices”
Lines 100-109: the paragraph is written as a summary of the study. It is necessary to clearly formulate the purpose and objectives of this study
References
Line 481: “Pacheco Muñoz” need change to “Pacheco-Muñoz”
Line 500: “Osorio Olvera” need change to “Osorio-Olvera”
Author Response
Abstract
In my opinion, it is necessary to describe in more detail not only what has been done, but also the results obtained and the patterns identified, as, for example, it is written in Simple Summary.
>>Thank you for your suggestion. We have revised the abstract according to your suggestion.
lines 86-87: “However, another study found that phylogenetic diversity increased with increasing building area, impervious surface, and pedestrian flow density” [need references].
>>Thank you for your suggestion. We have added a reference after this sentence.
lines 90-91: “As an important component of urban habitats, university campuses host abundant biodiversity globally”. The use of the word "globally" raises doubts. Need references too.
>>Thank you for your suggestion. After careful consideration, we decided to delete the word "globally" and a reference is also added to justify.
Line 96: “the Shannon-Wiener and Simpson indices” need corrected to “the Shannon-Wiener and Simpson biodiversity indices”.
>>Done.
Lines 100-109: the paragraph is written as a summary of the study. It is necessary to clearly formulate the purpose and objectives of this study.
>>Thank you for the nice suggestion. We have briefly summarized the purpose of this study in the revised version.
References
Line 481: “Pacheco Muñoz” need change to “Pacheco-Muñoz”.
>>changed.
Line 500: “Osorio Olvera” need change to “Osorio-Olvera”.
>>changed.